# Investigation of High-Efficiency Iterative ILU Preconditioner Algorithm for Partial-Differential Equation Systems

**Yan-Hong Fan** [1] , **Ling-Hui Wang** [1],*, **You Jia** [1] , **Xing-Guo Li** [1] , **Xue-Xia Yang** [1] **and Chih-Cheng Chen** [2],*

[1] Department of Mechanics, School of Applied Science, Taiyuan University of Science and Technology, Taiyuan 030024, China; yhfan0509@tyust.edu.cn (Y.-H.F.); YouJia@tyust.edu.cn (Y.J.); XingGuoLi@tyust.edu.cn (X.-G.L.); XueXiaYang@tyust.edu.cn (X.-X.Y.)
[2] Information and Engineering College, Jimei University, Xiamen 361021, Fujian, China
\* Correspondence: LingHuiWang@tyust.edu.cn (L.-H.W.); 201761000018@jmu.edu.cn (C.-C.C.)

**Abstract:** In this paper, we investigate an iterative incomplete lower and upper (ILU) factorization preconditioner for partial-differential equation systems. We discretize the partial-differential equations into linear equation systems. An iterative scheme of linear systems is used. The ILU preconditioners of linear systems are performed on the different computation nodes of multi-central processing unit (CPU) cores. Firstly, the preconditioner of general tridiagonal matrix equations is tested on supercomputers. Then, the effects of partial-differential equation systems on the speedup of parallel multiprocessors are examined. The numerical results estimate that the parallel efficiency is higher than in other algorithms.

**Keywords:** iterative ILU; preconditioner; partial-differential equations; parallel computation

## 1. Introduction

In applied sciences, such as computational electromagnetics, the solving of partial-differential equation systems is usually touched upon. Many variables need to be sought for solving engineering problems. These often need to be transformed into a solution of partial differential equations. When solving partial differential equations, the equations need to be discretized. When discretizing partial differential equations, symmetric systems of equations are usually gotten. Hence, it is necessary to use the idea of symmetry to solve partial differential equations. Several studies on multi-computers have appeared. For instance, Eric Polizzi and Ahmed H. Sameh [1] contributed a spike algorithm as a parallel solution to hybrid banded equations. The algorithm firstly decomposes banded equations into block-tridiagonal form and then makes full use of the divide and conquer technique. However, by increasing the bandwidth, the parallel computation becomes much more complex, leading to a decrease in the parallel efficiency. Obviously, the highly efficient parallelism of banded systems is of great importance. Methods for block-tridiagonal linear equations contain iterative algorithms such as the multi-splitting algorithm [2,3]. The multi-splitting algorithm (MPA) [2] can be used to solve large band linear systems of equations; however, it sometimes has lower parallel efficiency. In [4], a method for working out block-tridiagonal equations is provided by the authors. Any incomplete type preconditioner will be appropriate for the algorithm. Based on the Galerkin principle, the parallelism solution for large-scale banded equations is investigated in [5]. In [6], a parallel direct algorithm is used on multi-computers. In [7], a parallel direct method for large banded equations is presented. A preconditioner of large-scale banded equations is discovered in [8–14]. The block successive

over-relaxation method (BSOR) [10] can be adopted to solve large-scale systems of equations, but different parallel efficiencies will be presented because of the different optimal relaxation factors. These algorithms use parallelism to solve banded equations but they cannot contain solving partial differential equations. From better provision of a computing environment, a highly efficient preconditioner can be carried out on multi-computers [15–23]. Simultaneously, Krylov subspace solvers [24–30] and preconditioners [31–38] for large-scale banded equations are commonly used, including the generalized minimal residual (GMRES) [39]. The pseudo-elimination method with parameter k (PEk) [40] can be applied on multi-processors; however, the setting of parameter k will influent the speedup and parallel efficiency. These are mostly preconditioners for sparse linear systems or partial differential equation problems in Graphics Processing Unit (GPU) computation. However, these methods consume great computational effort. The development of a new algorithm which needs less calculation among every iteration and has more speedup and higher parallel efficiency is required. This paper is based on the symmetry subject of solving partial differential equation systems. The systems of equations are usually symmetric. In the process of solving them, the systems of equations need to be divided into blocks. The block equations may be symmetric or asymmetric, so this paper considers the general form of block equations. Of course, for symmetric block equations, the incomplete lower and upper factorization preconditioner (ILUP) algorithm is suitable. This paper is concerned with partial-differential equation systems of the form $Ax = b$. The associated iterative form $Mx^{(k+1)} = Nx^{(k)} + b$ is used. The linear tridiagonal special form is tested on multi-processors. Then, the iterative ILUP for partial differential equation systems is used to examine multi central processing unit (CPU) cores.

The outline is as mentioned hereunder. Section 2 describes a decomposition strategy of a parallel algorithm. Section 3 documents the analysis of convergence. Section 4 introduces the parallel implementation of this algorithm. The analysis of results computations with numerical examples including a large-scale system of equations and partial-differential equations are presented in Section 5. Finally, we conclude the paper in Section 6.

## 2. Decomposition Strategy

Consider large-scale band equations

$$Ax = b \tag{1}$$

that is

$$
\begin{pmatrix}
A_1 & B_1 & & & & \\
C_2 & A_2 & B_2 & & & \\
& \ddots & \ddots & \ddots & & \\
& & C_{n-1} & A_{n-1} & B_{n-1} \\
& & & C_n & A_n
\end{pmatrix}
\begin{pmatrix}
x_1 \\ x_2 \\ \vdots \\ x_{n-1} \\ x_n
\end{pmatrix}
=
\begin{pmatrix}
b_1 \\ b_2 \\ \vdots \\ b_{n-1} \\ b_n
\end{pmatrix}
$$

where $A_i$, $B_i$, and $C_i$ are $d_i \times d_i$, $d_i \times d_{i+1}$, and $d_i \times d_{i-1}$, and $x_i$, $b_i$ are the $d_i-$ vectors of the unknowns and the right–hand side,

The coefficient matrix $A$ can be approximately decomposed as

$$A \approx GH \tag{2}$$

Generally, supposing $n = pm(m \geq 2, m \in \mathbf{Z})$, where p represents the processors, let

$$M = GH$$

where

$$
G = \begin{pmatrix}
I_1 & & & & & & & & & & & & & \\
L_2 & I_2 & & & & & & & & & & & & \\
& \ddots & \ddots & & & & & & & & & & & \\
& & L_m & I_m & S_m & & & & & & & & & \\
& & & & I_{m+1} & & & & & & & & & \\
& & & & L_{m+2} & I_{m+2} & & & & & & & & \\
& & & & & & \ddots & \ddots & & & & & & \\
& & & & & & & L_{2m} & I_{2m} & S_{2m} & & & & \\
& & & & & & & & & I_{2m+1} & & & & \\
& & & & & & & & & L_{2m+2} & I_{2m+2} & & & \\
& & & & & & & & & & & \ddots & \ddots & \\
& & & & & & & & & & & & L_n & I_n
\end{pmatrix}
$$

$$
H = \begin{pmatrix}
U_1 & S_1 & & & & & & & & & & \\
& \ddots & \ddots & & & & & & & & & \\
& & U_{m-1} & S_{m-1} & & & & & & & & \\
& & & U_m & & & & & & & & \\
& & L_{m+1} & U_{m+1} & S_{m+1} & & & & & & & \\
& & & & \ddots & \ddots & & & & & & \\
& & & & & U_{2m-1} & S_{2m-1} & & & & & \\
& & & & & & U_{2m} & & & & & \\
& & & & & L_{2m+1} & U_{2m+1} & S_{2m+1} & & & & \\
& & & & & & & & \ddots & \ddots & & \\
& & & & & & & & & U_{n-1} & S_{n-1} & \\
& & & & & & & & & & U_n &
\end{pmatrix}
$$

$$(3)$$

in which

$S_i = B_i,\ i = m(q-1)+1, \cdots, m(q-1)+m-1,\ q = 1, \cdots, p$

$S_i = B_i A_{i+1}^{-1},\ i = mq,\ q = 1, 2 \cdots, p-1$

$L_i = C_i,\ i = m(q-1)+1,\ q = 2, 3, \cdots, p$

$L_i = C_i U_{i-1}^{-1},\ i = m(q-1)+2, \cdots, m(q-1)+m,\ q = 1, \cdots, p$

$U_i = A_i,\ i = m(q-1)+1,\ q = 1, \cdots, p$

$U_i = A_i - L_i S_{i-1},\ i = mq+2, \cdots, mq+m-1,\ q = 0, \cdots, p-1;\ i = mq+m, q = p-1$

$U_i = A_i - L_i S_{i-1} - S_i L_{i+1},\ i = m(q-1)+m,\ q = 1, \cdots, p-1$

and $I_i$ is a $d_i \times d_i$ unit matrix, $i = 1, \cdots, n$.

Then

$$N = M - A$$

that is

$$
N = \begin{pmatrix}
(O) & & & & & & & & \\
& & & O & S_m S_{m+1} & & & & \\
& & O & & & & & & \\
& L_{m+2}L_{m+1} & & & & & & & \\
& & & & (O) & & & & \\
& & & & & O & S_{2m}S_{2m+1} & & \\
& & & & & & \ddots & & \\
& & & & & & & O & \\
& & & & & & L_{(p-1)m+2}L_{(p-1)m+1} & & \\
& & & & & & & & (O)
\end{pmatrix}
$$

where $(O)$ is the $\sum_{i=1}^{m} d_i \times \sum_{i=1}^{m} d_i$ zero matrix. Therefore, the new iterative scheme for the large-scale band system of equations is

$$GHx^{(k+1)} = Nx^{(k)} + b \tag{4}$$

where the iterative matrix is

$$T = H^{-1}G^{-1}N$$

Obviously, $GH$ is nonsingular, which is the necessary condition that the algorithm holds. In terms of the structure of $G$ and $H$, the parallelism of the iterative algorithm is preferable.

The strategy is an ILUP algorithm. Compared with published algorithms [2,10,40], the ILUP algorithm requires less multiplication and adds calculation among every iteration, meaning this algorithm has more speedup and higher parallel efficiency. It is appropriate for solving the large-scale system of equations and partial-differential equations for multi-core processors.

## 3. Analysis of Convergence

### 3.1. Preliminary

Here, some notations are introduced. Two definitions and one lemma are mentioned.

**Definition 1.** *([39]) A real $n \times n$ matrix $A = (a_{i,j})$ with $a_{i,j} \leq 0$ for all $i \neq j$ is an M-matrix if $A$ is nonsingular and $A^{-1} \geq O$.*

**Definition 2.** *([39]) The matrix $A, M, N, A = M - N$ is a regular splitting of $A$ if $M$ is nonsingular, $M^{-1} \geq O$, $N \geq O$.*

**Lemma 1.** *([39]) Presume $A = M - N$ is a regular splitting of $A$. Then, $A$ is nonsingular and $A^{-1} \geq O$, if and only if $\rho(M^{-1}N) < 1$.*

### 3.2. Proposition and Theorem

Note that the inverse matrix of the following matrix is gained by the algorithm of the Gaussian elimination. Firstly, from the definitions and lemma, a proposition is obtained as follows.

**Proposition 1.** *If $A$ is an M-matrix, in this way, the matrices $U_i$ $(i = 1, 2, 3, \cdots, n)$ defined by Expression (3) satisfy $U_i^{-1} \geq O$.*

**Proof.** From Expression (3), in terms of the contracture of $A$, $G$, $H$ and $M = GH$, $N = M - A$, we have

$U_i = A_i$, $i = m(q-1)+1$, $q = 1, \cdots, p$
$U_i = A_i - L_i S_{i-1} = A_i - C_i U_{i-1}^{-1} B_{i-1}$, $i = m(q-1)+2, \cdots, m(q-1)+m-1$, $q = 1, \cdots, p-1$;
$i = n - m + 1, \cdots, n$, $q = p$;
$U_i = A_i - L_i S_{i-1} - S_i L_{i+1}$, $i = m(q-1)+m$, $q = 1, \cdots, p-1$.

As $A$ is an M-matrix, then $U_i^{-1} \geq O$ for $i = m(q-1)+1$, $q = 1, \cdots, p$

$$Let\ W_i = \begin{pmatrix} A_{(i-1)m+1} & B_{(i-1)m+1} & & \\ C_{(i-1)m+2} & A_{(i-1)m+2} & \ddots & \\ & \ddots & \ddots & B_{im-1} \\ & & C_{im} & A_{im} \end{pmatrix}, then\ W_i^{-1} \geq O.$$

Since the block on the $m$-th row and $m$-th column of $W_i^{-1}$ is $U_i^{-1}$ for $i = m(q-1)+2, \cdots, m(q-1)+m-1$ and $q = 1, \cdots, p-1$;

Hence, $U_i^{-1} \geq O$ for $i = m(q-1)+2, \cdots, m(q-1)+m-1$ and $q = 1, \cdots, p-1$;

Furthermore,

$$V_i = \begin{pmatrix} A_{(i-1)m+1} & B_{(i-1)m+1} & & \\ C_{(i-1)m+2} & A_{(i-1)m+2} & \ddots & \\ & \ddots & \ddots & B_{(i-1)m+m} \\ & & C_{(i-1)m+m+1} & A_{(i-1)m+m+1} \end{pmatrix},$$

Similarly, the block on the $m$-th row and $m$-th column of $V_i^{-1}$ is $U_i^{-1}$ for $i = m(q-1)+m$, $q = 1, \cdots, p-1$ by inducing. Therefore, $U_i^{-1} \geq O$ for $i = m(q-1)+m$, $q = 1, \cdots, p-1$. Then, we have $U_i^{-1} \geq O$ $(i = 1, \cdots, n)$.

Secondly, taking advantage of the above lemma and proposition, a theorem is given. □

**Theorem 1.** *If $A$ is an M-matrix, then the approximate factorization of matrix $A$ can be represented by Expression (2), and the iterative scheme Algorithm (4) converges to $X^* = A^{-1}b$.*

**Proof.** From the above proposition, the approximate factorization of matrix $A$ can be represented by Expression (2).

Firstly, prove $N \geq O$.

As $A$ is an M-matrix, then $A_{im+1}^{-1} \geq O$, $B_{im+1} \leq O$, $B_{im} \leq O$, $C_{im+1} \leq O$, $C_{im+2} \leq O$, for $i = 1, \cdots, p-1$. Hence, $B_{im} A_{im+1}^{-1} B_{im+1} \geq O$, $C_{im+2} A_{im+1}^{-1} C_{im+1} \geq O$, for $i = 1, \cdots, p-1$. Therefore, $N \geq O$.

Secondly, prove $M^{-1} \geq O$.

$$Since\ M^{-1} = \tilde{U}^{-1} \tilde{L}^{-1}, provided\ \tilde{L}^{-1} = \begin{pmatrix} \widehat{L}_1 & -\widehat{S}_1 & & & \\ & \widehat{L}_2 & -\widehat{S}_2 & & \\ & & \ddots & \ddots & \\ & & & \widehat{L}_{p-1} & -\widehat{S}_{p-1} \\ & & & & \widehat{L}_p \end{pmatrix},$$

where

$$\widehat{L}_i = \begin{pmatrix} I_{(i-1)m+1} & & & \\ -L_{(i-1)m+2} & I_{(i-1)m+2} & & \\ & \ddots & \ddots & \\ & & -L_{im} & I_{im} \end{pmatrix}, i = 1, \cdots, p, \quad -\widehat{S}_i = \begin{pmatrix} O \\ \vdots \\ O \\ -S_{im} \end{pmatrix}, i = 1, \cdots, p-1,$$

and

$$\tilde{U}^{-1} = \begin{pmatrix} \widehat{U}_1 & & & & \\ \widehat{C}_2 & \widehat{U}_2 & & & \\ & \ddots & \ddots & & \\ & & \widehat{C}_{p-1} & \widehat{U}_{p-1} & \\ & & & \widehat{C}_p & \widehat{U}_p \end{pmatrix},$$

where

$$\widehat{U}_i = \begin{pmatrix} U^{-1}_{(i-1)m+1} & -U^{-1}_{(i-1)m+1}B_{(i-1)m+1}U^{-1}_{(i-1)m+2} & \cdots & (-1)^{m-1}\prod_{j=1}^{m-1} U^{-1}_j B_j U^{-1}_{im} \\ & \ddots & \ddots & \vdots \\ & & U^{-1}_{(i-1)m+m-1} & -U^{-1}_{(i-1)m+m-1}B_{(i-1)m+m-1}U^{-1}_{im} \\ & & & U^{-1}_{im} \end{pmatrix},$$

$$\widehat{C}_i = \begin{pmatrix} -C_{(i-1)m+1}U^{-1}_{(i-1)m} \\ \vdots \\ O \\ O \end{pmatrix}. \ i = 2, \cdots, p.$$

According to the proposition, $U_i^{-1} \geq O(i = 1, \cdots, n)$. Since

$$L_{(i-1)m+j} = C_{(i-1)m+j}U^{-1}_{(i-1)m+j-1}, j = 2, \cdots, m, i = 1, \cdots, p$$

we have $-L_{(i-1)m+j} \geq O, i = 1, \cdots, p, j = 2, \cdots, m$. Therefore, $\tilde{L}^{-1} \geq O, \tilde{U}^{-1} \geq O M^{-1} \geq O$.

Finally, based on $M^{-1} \geq O, N \geq O$ and Lemma 1, we conclude that $\rho(M^{-1}N) < 1$. That is, this algorithm converges. □

This section shows that the condition in the theorem is a sufficient condition for convergence of the algorithm. If $A$ is not an M-matrix, Algorithm (4) is sometimes convergent, as is shown in the following section (Example 1).

## 4. Parallel Implementations

### 4.1. Storage Method

For the $i$-th processor $P_i(i = 1, \cdots, p)$, allocate $A_{(i-1)m+j}$, $B_{(i-1)m+j}$, $C_{(i-1)m+j}$ ($i \neq p, j = 1, \cdots, m, m + 1; i = p, j = 1, \cdots, m$), $b_{(i-1)m+j}$ ($j = 1, \cdots, m$), the initial vector $x^{(0)}_{(i-1)m+j}$, and the convergence tolerance $\varepsilon$.

### 4.2. Circulating

(1) $Gy = b + Nx^{(k)}$ is solved to obtain $y$.

$P_i$ ($i = 1, \cdots, p - 1$) acquires $x^{(k)}_{(i+1)m+2}$ from $P_{i+1}$ and then computes to obtain $y_{(i-1)m+q}, q = 1, \cdots, m - 1, i = 1, \cdots, p$ and $y_n$. $P_i$ ($i = 1, \cdots, p - 1$) gains $y_{(i+1)m+1}$ from $P_{i+1}$ and then obtains $y_{im}, i = 1, \cdots, p - 1$.

(2) $Hx^{(k+1)} = y$ is solved to obtain $x^{(k+1)}$.

$P_i$ ($i = 1, \cdots, p$) computes to obtain $x^{(k+1)}_{(i-1)m+q}$ ($q = 2, \cdots, m, i = 1, \cdots, p$) and $x^{(k+1)}_1$. The -$i$th processor $P_i$ ($i = 2, \cdots, p$) gains $x^{(k+1)}_{im}$ from $P_{i-1}$ and then computes to obtain $x^{(k+1)}_{(i-1)m+1}, i = 2, \cdots, p$.

(3) On $P_i(i = 1, \cdots, p)$, judge $\|x_{(i-1)m+j}^{(k+1)} - x_{(i-1)m+j}^{(k)}\| \leq \varepsilon$. Following this, stop if correct, or otherwise, go back to step (1).

## 5. Results Analysis of Numerical Examples

For testing the new algorithm, some results on the Inspur TS10000 cluster have been given by the new algorithm and order 2 multi-splitting algorithm [2], which is a well-known parallel iterative algorithm. The PEk method [40] is used on the inner iteration of the order 2 multi-splitting algorithm. Suppose $d_i = d_{i-1} = d_{i+1} = t$, $x_i^{(0)} = (0, \cdots, 0)_{t \times 1}^{\mathrm{T}}$, $\|x^{(k+1)} - x^{(k)}\|_\infty < \bar{\varepsilon}$, $\bar{\varepsilon} = 10^{-10}$.

In the tables, P is the number of processors, l is the inner iteration time, $k$ is the parameter of the PEk method, T is the run time (in seconds), I is the iterative time, S is the speedup and E is the parallel efficiency (E = S/P). In the following figures, ILUP, BSOR, PEk, and MPA, respectively, denote the iterative incomplete lower and upper factorization preconditioner, the block successive over-relaxation method, the PEk method, and the multi-splitting algorithm.

### 5.1. Results Analysis of the Large-Scale System of Equations

**Example 1.** *A in Expression (1) represents*

$$A_i = \begin{bmatrix} 12 & -2 & & & \\ -3 & 12 & -2 & & \\ & \ddots & \ddots & \ddots & \\ & & -3 & 12 & -2 \\ & & & -3 & 12 \end{bmatrix}_{t \times t}, B_i = \begin{bmatrix} 2.2 & -1.3 & & & \\ -3 & 2.2 & -1.3 & & \\ & \ddots & \ddots & \ddots & \\ & & -3 & 2.2 & -1.3 \\ & & & -3 & 2.2 \end{bmatrix}_{t \times t},$$

$$C_i = \begin{bmatrix} 2 & 2 & & & \\ -1 & 2 & 2 & & \\ & \ddots & \ddots & \ddots & \\ & & -1 & 2 & 2 \\ & & & -1 & 2 \end{bmatrix}_{t \times t}, b_i = \begin{bmatrix} (i-1)k+1 \\ (i-1)k+2 \\ \vdots \\ ik-1 \\ ik \end{bmatrix}_{t \times 1 \ldots}, \text{ and } (i = 1, 2, \cdots, n),$$

*where $B_n = C_1 = O$, $n = 300$, and $t = 300$. The numerical results are shown in Tables 1–5, and in Figures 1 and 2.*

The first example is not a numerical simulation regarding any partial differential equations (PDE); we use this example in order to test the correctness of the iterative incomplete lower and upper factorization preconditioner algorithm. The first example can build a good foundation for the second example regarding PDE. The solutions to the large-scale system of equations for Example 1 by the ILUP are shown in Table 1 and the details of these are as follows: This problem requires solving with more than eight processors and the number of iterations is 238. When increasing the number of processors, time and parallel efficiency all decrease. The number of processors for solving Example 1 transforms from 4 to 64 and the parallel efficiency changes from 91.14% to 73.80%. All of the parallel efficiency values are higher than those in published works, including Cui et al.'s [10], Zhang et al.'s [40], and Yun et al.'s [2] methods, with the values being above 73%. No matter how many processors are used to calculate the problem, the error tolerance of this example is the same: $6.897 \times 10^{-11}$.

**Table 1.** The iterative incomplete lower and upper factorization preconditioner (ILUP) for Example 1.

| P | 1 | 4 | 8 | 16 | 32 | 64 |
|---|---|---|---|---|---|---|
| T | 119.1036 | 32.6697 | 17.5870 | 9.6202 | 4.8371 | 2.5217 |
| I | 233 | 237 | 238 | 238 | 238 | 238 |
| S | | 3.6457 | 6.7723 | 12.3806 | 24.6231 | 47.2324 |
| E | | 0.9114 | 0.8465 | 0.7738 | 0.7695 | 0.7380 |
| Δ | $6.897 \times 10^{-11}$ | $6.897 \times 10^{-11}$ | $6.897 \times 10^{-11}$ | $6.897 \times 10^{-11}$ | $6.897 \times 10^{-11}$ | $6.897 \times 10^{-11}$ |

The results of Example 1 when using the BSOR method [10] are listed in Table 2. When more than four processors are used to resolve the problem of Example 1, the number of iterations is 216. When increasing the number of processors, the time and parallel efficiency decrease. The cost of the time of every iteration and communication is more than that found when using the ILUP algorithm for the large-scale system of equations. Hence, the speedup, which is less than that found when using the ILUP algorithm, decreases. Thus the parallel efficiency is not better than that found when using the ILUP algorithm for the large-scale system of equations. When the number of processors for solving Example 1 is four, the parallel efficiency is 59.56%; however, the parallel efficiency is 91.14% for four processors when using the ILUP algorithm. When increasing the number of processors, the parallel efficiency decreases to 44.81%, which is lower than that found when using the ILUP algorithm.

**Table 2.** The key to the block successive over relaxation method (BSOR) method for Example 1 ($\omega = 2.0$).

| P | 1 | 4 | 8 | 16 | 32 | 64 |
|---|---|---|---|---|---|---|
| T | 112.0383 | 47.0284 | 25.0183 | 14.0130 | 7.5833 | 3.9065 |
| I | 211 | 216 | 216 | 216 | 216 | 216 |
| S | | 2.3824 | 4.4783 | 7.9953 | 14.7743 | 28.6800 |
| E | | 0.5956 | 0.5598 | 0.4997 | 0.4617 | 0.4481 |

The results of Example 1 when using the PEk method published by Zhang et al. [40] are described as Table 3. When more than four processors are used to resolve the problem of Example 1, the number of iterations is 227. When increasing the number of processors, the time and parallel efficiency decrease. The cost of the time of every iteration and communication is more than that when using the ILUP algorithm for the large-scale system of equations. Hence, the speedup, which is less than that found when using the ILUP algorithm, decreases. Therefore, the parallel efficiency is poorer than that found when using the ILUP algorithm for the large-scale system of equations. When the number of processors used when solving Example 1 is four, the parallel efficiency is 64.08%; however, the parallel efficiency is 91.14% for four processors when using the ILUP algorithm. When increasing the number of processors, the parallel efficiency decreases to 44.79%, corresponding to the parallel efficiency when using the BSOR method, which is lower than that found when using the ILUP algorithm, 73.80%.

**Table 3.** Answers for the pseudo-elimination method with parameter k (PEk) for Example 1 (k = 1.6).

| P | 1 | 4 | 8 | 16 | 32 | 64 |
|---|---|---|---|---|---|---|
| T | 114.3098 | 44.5992 | 24.7489 | 14.2286 | 7.6159 | 3.9878 |
| I | 224 | 227 | 227 | 227 | 227 | 227 |
| S | | 2.5630 | 4.6188 | 8.0338 | 15.0094 | 28.6649 |
| E | | 0.6408 | 0.5773 | 0.5021 | 0.4690 | 0.4479 |

The results of Example 1 when using the multi-splitting algorithm (MPA) published by Yun et al. [2] are introduced in Table 4. As seen in Table 4, when more than four processors are used to solve the problem of Example 1, the number of iterations is 174. When increasing the number of processors, the time and parallel efficiency decrease. The cost of the time of every iteration and communication is

more than that when using the ILUP algorithm for the large-scale system of equations. Hence, the speedup, which is less than that found when using the ILUP algorithm, decreases. Thus, the parallel efficiency is poorer than when using the ILUP algorithm for the large-scale system of equations. When the number of processors for solving Example 1 is four, the parallel efficiency is 55.64%, 33.50% less than that that found when using the ILUP algorithm. When increasing the number of processors, the parallel efficiency decreases to 40.82%, about 4% less than the parallel efficiency obtained with the BSOR method, which is 23% lower than that that found when using the ILUP algorithm.

**Table 4.** The solutions to the multi-splitting algorithm (MPA) used for Example 1.

| P | 1 | 4 | 8 | 16 | 32 | 64 |
|---|---|---|---|---|---|---|
| T | 103.597 | 46.547 | 24.716 | 13.717 | 7.472 | 3.9564 |
| I | 172 | 174 | 174 | 174 | 174 | 174 |
| S | | 2.2256 | 4.1915 | 7.5525 | 13.8647 | 26.1254 |
| E | | 0.5564 | 0.5239 | 0.4720 | 0.4333 | 0.4082 |

This section compares the speedup and parallel efficiency performance of the ILUP algorithm with methods in other recently published works, including Cui et al.'s [10], Zhang et al.'s [40], and Yun et al.'s [2] methods. Table 5 introduces a summary and comparison of the speedup and parallel efficiency with the different methods used for Example 1 on 64 CPU cores, which is better than other works [2,10,40]. As seen in Table 5, the speedup obtained with our method for Example 1 on 64 CPU cores is 47.2324, and the parallel efficiency is 73.80%. The parallel efficiency obtained with the ILUP algorithm is about 29% higher than that obtained using the BSOR method. The parallel efficiency is 29.01% more than that obtained using the PEk method. The parallel efficiency obtained with the BSOR method corresponds to the parallel efficiency obtained with the PEk method. The parallel efficiency is 23% higher than that obtained using the MPA algorithm.

**Table 5.** Comparison speedup and parallel efficiency with the different methods used for Example 1 on 64 central processing unit (CPU) cores.

| Compared List | ILUP Algorithm | Block Successive over Relaxation Method [10] | Pseudo-Elimination Method with Parameter k [40] | Multi-Splitting Algorithm [2] |
|---|---|---|---|---|
| Speedup | 47.2324 | 28.6800 | 28.6649 | 26.1254 |
| Parallel Efficiency | 0.7380 | 0.4481 | 0.4479 | 0.4082 |

Figure 1 illustrates the speedup performances obtained with the ILUP algorithm and the other three methods for Example 1 at different CPU cores. As seen from Figure 1, when increasing the number of processors, the speedup obtained using all the methods increases. No matter how great the number of processors, the speedup obtained using the ILUP algorithm is significantly higher than that obtained using the other three methods, especially when the number of processors is more. Regardless of the number of processors, the speedup values obtained using the BSOR method, the PEk method, and the MPA algorithm are close, particularly those obtained with the BSOR method and the PEk method.

Figure 2 shows the parallel efficiency performance of the ILUP algorithm and the other three methods for Example 1 at different CPU cores. As seen from Figure 2, when increasing the number of processors, the parallel efficiency obtained using all the methods decreases. Regardless of the number of processors, the parallel efficiency obtained using the ILUP algorithm is much higher than that found using the other three methods, maintaining a value of more than 70%. No matter the number of processors, the parallel efficiency values obtained using the PEk method, the BSOR method, and the MPA algorithm are lower and nearer, especially those found using the BSOR method and the PEk method. In particular, when the number of processors is 64, the parallel efficiency obtained

using the ILUP algorithm rises above 73%; however, the parallel efficiencies obtained using the BSOR method, the PEk method, and the MPA algorithm are only about 40%. The ILUP algorithm has the clear superiority of producing exceedingly higher parallel efficiency values.

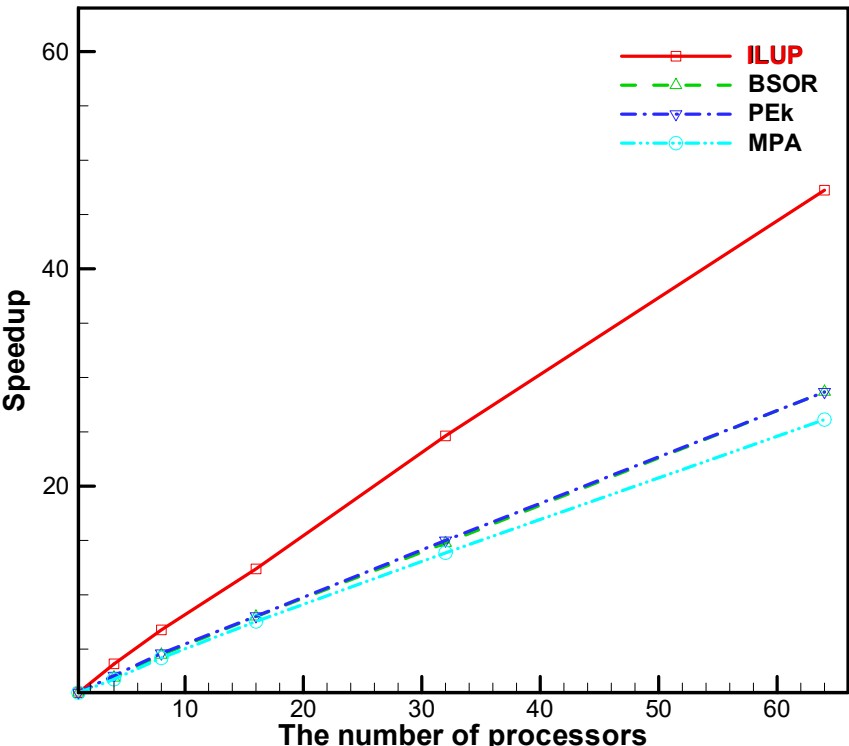

**Figure 1.** The speedup values for Example 1.

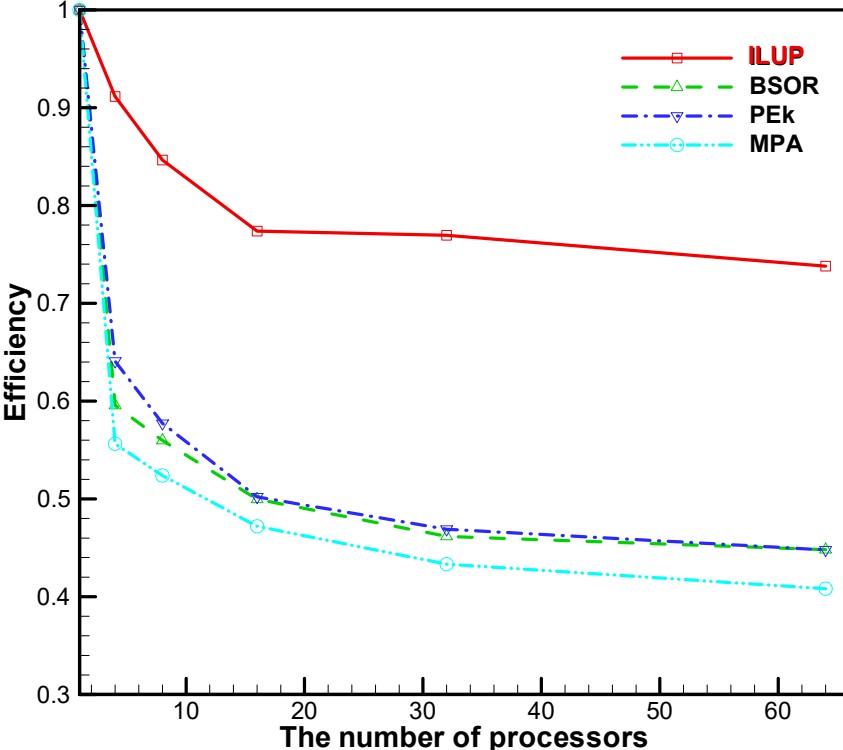

**Figure 2.** The parallel efficiency values for Example 1.

## 5.2. Results Analysis of the Partial-Differential Equations

**Example 2.** *Given the equations*

$$C_x \frac{\partial^2 u}{\partial x^2} + C_y \frac{\partial^2 u}{\partial y^2} + (C_1 \sin 2\pi x + C_2) \frac{\partial u}{\partial x} + (D_1 \sin 2\pi y + D_2) \frac{\partial u}{\partial y} + Eu = 0$$

$0 \leq x \leq 1, \ 0 \leq y \leq 1$

*and*

$u\big|_{x=0} = u\big|_{x=1} = 10 + \cos \pi y$

$u\big|_{y=0} = u\big|_{y=1} = 10 + \cos \pi x,$

$C_x, C_y, C_1, C_2, D_1, D_2$ *and* $E$ *are invariants. Let* $C_x = C_y = E = 1, C_1 = C_2 = D_1 = D_2 = 0$ *and* $h = 1/101$. *The results are given in Tables 6–10 and in Figures 3 and 4.*

The finite difference method is used to discretize Example 2 in the tests. We adopt second-order central difference schemes to discretize Example 2 and then converse the format for numerical simulation; lastly, we test the iterative incomplete lower and upper factorization preconditioner algorithm on different processors. The results to the partial-differential equations for Example 2 obtained using the ILUP are listed in Table 6. The details are thus: This problem was solved with more than four CPU cores and the number of iterations was 560. When increasing the number of processors, the time and the parallel efficiency can be seen to all decrease. When the number of processors used for solving Example 2 changes from 4 to 64 the parallel efficiency changes from 89.48% to 71.64%. All of the parallel efficiency values are higher than in the published works [2,10,40], being above 71%. Regardless of how many processors are used to compute Example 2, the error allowance of this problem can be seen to be equally $3.158 \times 10^{-11}$.

**Table 6.** The iterative incomplete lower and upper factorization preconditioner for Example 2.

| P | 1 | 4 | 8 | 16 | 32 | 64 |
|---|---|---|---|---|---|---|
| T | 121.7960 | 34.0280 | 19.6270 | 10.2140 | 5.1830 | 2.6565 |
| I | 578 | 560 | 560 | 560 | 560 | 560 |
| S | | 3.5793 | 6.2055 | 11.9244 | 23.4991 | 45.8483 |
| E | | 0.8948 | 0.7757 | 0.7453 | 0.7343 | 0.7164 |
| Δ | $3.158 \times 10^{-10}$ | $3.158 \times 10^{-10}$ | $3.158 \times 10^{-10}$ | $3.158 \times 10^{-10}$ | $3.158 \times 10^{-10}$ | $3.158 \times 10^{-10}$ |

The results for Example 2 obtained with the BSOR method [10] are listed in Table 7. When more than four processors are used to resolve the problem of Example 2, the number of iterations is 793. When increasing the number of processors, the time and parallel efficiency decrease. The cost of the time of every iteration and communication is more than that obtained using the ILUP algorithm for the large-scale system of equations. Hence, the speedup, which is less than that found when using the ILUP algorithm, decreases. Thus, the parallel efficiency is not as good as that found using the ILUP algorithm for the partial-differential equations. When the number of processors used for solving Example 2 is four, the parallel efficiency is 86.24%, 3.24% lower than that found when using the ILUP algorithm for the partial-differential equations. With increasing the number of processors, the parallel efficiency decreases to 52.42%, which is less than that obtained using the ILUP algorithm, 71.64%.

**Table 7.** The key to the BSOR method for Example 2 ($\omega = 2.0$).

| P | 1 | 4 | 8 | 16 | 32 | 64 |
|---|---|---|---|---|---|---|
| T | 144.8230 | 41.9830 | 26.6220 | 14.1590 | 7.6370 | 4.3165 |
| I | 779 | 793 | 793 | 793 | 793 | 793 |
| S | | 3.4496 | 5.4400 | 10.2283 | 18.9633 | 33.5510 |
| E | | 0.8624 | 0.6800 | 0.6393 | 0.5926 | 0.5242 |

The results obtained for Example 2 using the PEk method [40] are given in Table 8. When more than four processors are used to resolve the problem of Example 2, the number of iterations is 798. When increasing the number of processors, the time and parallel efficiency decrease. The cost of the time of every iteration and communication is more than that obtained using the ILUP algorithm for the large-scale system of equations. Hence, the speedup, which is less than that obtained when using the ILUP algorithm, decreases. Thus, the parallel efficiency is poorer than that found when using the ILUP algorithm for the partial-differential equations. When the number of processors used for solving Example 2 is four, the parallel efficiency is 80.59%, which is 8.89% lower than that found when using the ILUP algorithm. When increasing the number of processors, the parallel efficiency decreases to 48.40%, which is 23.24% lower than that obtained with the ILUP algorithm.

**Table 8.** Answers to the PEk method for Example 2 (k = 2.7).

| P | 1 | 4 | 8 | 16 | 32 | 64 |
|---|---|---|---|---|---|---|
| T | 157.7210 | 48.9280 | 29.4860 | 16.0790 | 9.3640 | 5.0917 |
| I | 786 | 798 | 798 | 798 | 798 | 798 |
| S | | 3.2235 | 5.3490 | 9.8091 | 16.8433 | 30.9764 |
| E | | 0.8059 | 0.6686 | 0.6131 | 0.5264 | 0.4840 |

The results for Example 2 obtained with the multi-splitting algorithm [2] are introduced in Table 9. As seen in Table 9, when more than four processors are used to solve the problem of Example 2, the number of iterations is 838. When increasing the number of processors, the time and parallel efficiency decrease. The cost of the time of every iteration and communication is more than that found when using the ILUP algorithm for the partial-differential equations. Hence, the speedup, which is less than that found using the ILUP algorithm, decreases. Thus, the parallel efficiency is poorer than that obtained using the ILUP algorithm for the large-scale system of equations. When the number of processors used for solving Example 2 is four, the parallel efficiency is 78.25%, 11.23% less than that obtained using the ILUP algorithm. When increasing the number of processors, the parallel efficiency decreases to 46.34%, about 6% less than the parallel efficiency obtained with with the BSOR method, corresponding to the parallel efficiency obtained with the PEk technique, which is 25.3% lower than that found using the ILUP algorithm.

**Table 9.** The solutions to the multi-splitting algorithm for Example 2.

| P | 1 | 4 | 8 | 16 | 32 | 64 |
|---|---|---|---|---|---|---|
| T | 180.6459 | 57.7139 | 32.2524 | 17.7462 | 10.9967 | 6.0917 |
| I | 824 | 838 | 838 | 838 | 838 | 838 |
| S | | 3.1300 | 5.6010 | 10.1794 | 16.4273 | 29.6547 |
| E | | 0.7825 | 0.7001 | 0.6362 | 0.5134 | 0.4634 |

This section compares the speedup and parallel efficiency performance of the ILUP algorithm with methods in other recently published works, including Cui et al.'s [10], Zhang et al.'s [40], and Yun et al.'s [2] methods. Table 10 provides a summary and comparisons of speedup and parallel efficiency obtained using the different methods for Example 2 on 64 CPU cores, which is better than other published works. As seen in Table 10, the speedup in our method for Example 2 on 64 CPU cores is 45.8483 and the parallel efficiency is 71.64%. The parallel efficiency obtained using the ILUP algorithm is 19.22% higher than found using the BSOR method. The parallel efficiency is 23.24% more than that found using the PEk method. The parallel efficiency is 25.3% higher than that obtained using the MPA algorithm.

**Table 10.** Comparison of speedup and parallel efficiency values obtained using the different methods for Example 2 on 64 CPU cores.

| Compared List | ILUP Algorithm | Block Successive over Relaxation Method [10] | Pseudo-Elimination Method with Parameter k [40] | Multi-Splitting Algorithm [2] |
|---|---|---|---|---|
| Speedup | 45.8483 | 33.5510 | 30.9764 | 29.6547 |
| Parallel Efficiency | 0.7164 | 0.5242 | 0.4840 | 0.4634 |

Figure 3 compares the speedup performance of ILUP algorithm and the other three methods for Example 2 at different CPU cores. As seen from Figure 3, when increasing the number of processors, the speedup values of all the methods increase. Regardless of the number of processors, the speedup obtained using the ILUP algorithm is much higher than that found using the other three methods, in particular when the number of processors is greater. No matter the number of processors, the speedup values found using the BSOR method, the PEk method, and the MPA algorithm are close, especially for those found using the PEk technique and the MPA algorithm. For example, when the number of processors is 64, the speedup found using the ILUP algorithm rises above 45; however, the speedup values obtained using the BSOR method, the PEk method, and the MPA algorithm are only about 30. Obviously, the ILUP algorithm has the advantage of producing higher speedup values.

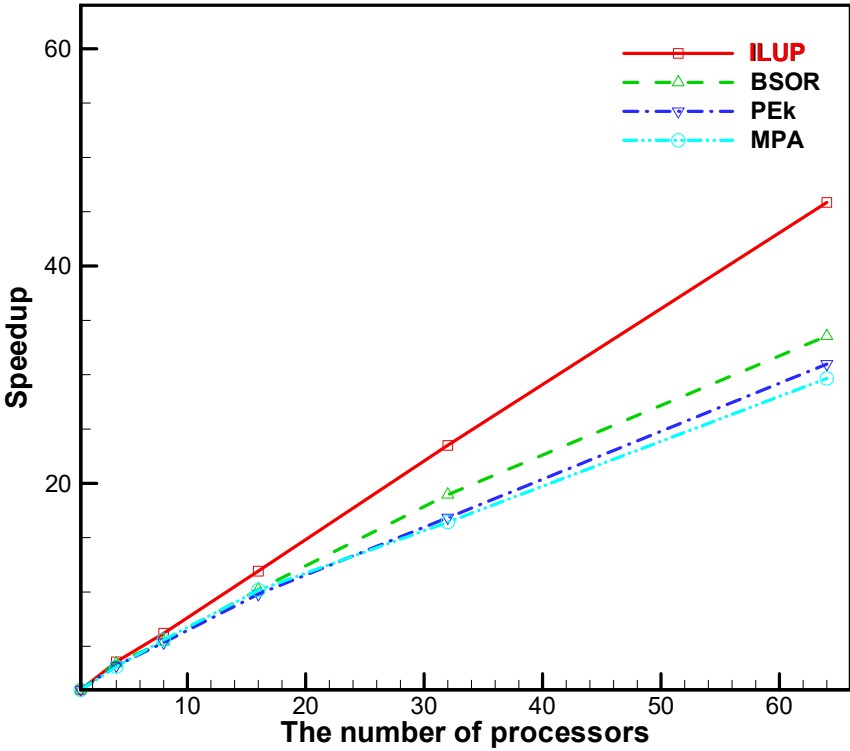

**Figure 3.** The speedup values for Example 2.

Figure 4 shows the parallel efficiency performance of the ILUP algorithm and the other three methods for Example 2 at different CPU cores. As seen from Figure 4, when increasing the number of processors, the parallel efficiency of all the methods decreases. Regardless of the number of processors, the parallel efficiency obtained using the ILUP algorithm is much higher than that found using the other three methods, maintaining a value of more than 70%. When increasing the number of processors, the parallel efficiency values obtained using the BSOR method, the PEk method, and the MPA algorithm are lower and sustain a descent, especially for those found using the MPA algorithm. In particular, when the number of processors is 64, the parallel efficiency obtained using the ILUP algorithm rises

above 71%; however, the parallel efficiency values found using the BSOR method, the PEk method, and the MPA algorithm are only about 50%. The ILUP algorithm is clearly beneficial in its production of exceedingly high parallel efficiency values.

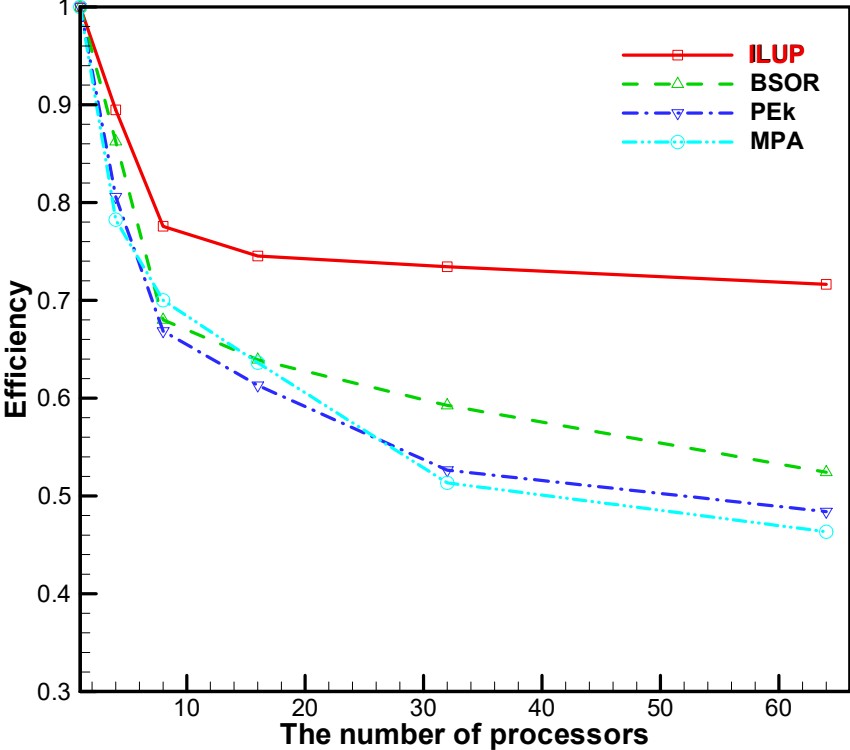

**Figure 4.** The parallel efficiency values for Example 2.

## 6. Conclusions

In this work, an iterative incomplete LU factorization preconditioner for partial-differential equation systems has been presented. The partial-differential equations were discretized into linear equations with the form Ax = b. An iterative scheme of linear systems was used. The iterative ILU preconditioners of linear systems and partial-differential equations systems were performed on different computation nodes of multi-CPU cores. From the above numerical results in the tables and figures, we can obtain the following conclusions:

1.  The ILUP algorithm for the large-scale system of equations and partial-differential equation systems was performed on different multi-CPU cores. The numerical results show that the solutions are consistent with the theory.
2.  From Example 1, when *A* is neither positive nor an M-matrix, the ILUP algorithm still converges.
3.  At any multi-CPU cores, the speedup of the ILUP algorithm for the system of equations is far higher than that found using the BSOR method [10], the PEk method [40], and the MPA algorithm [2]. Evidently, the ILUP algorithm has the advantage of producing higher speedup values.
4.  No matter the number of processors, the parallel efficiency of the ILUP algorithm is preferable. The parallel efficiency of the ILUP algorithm is higher than that of the other three algorithms. For example, the parallel efficiency of the ILUP algorithm achieves a value of above 73.8% (as seen in Table 5), which is higher than that for any other algorithm, including the BSOR method [10], the PEk method [40], and the MPA algorithm [2]. Obviously, the ILUP algorithm has the superiority of producing exceedingly high parallel efficiency values.

**Author Contributions:** Conceptualization, Y.-H.F. and L.-H.W.; methodology, L.-H.W.; software, Y.J.; validation, Y.J., X.-G.L., and X.-X.Y.; formal analysis, Y.-H.F.; investigation, Y.-H.F.; resources, L.-H.W.; data curation, Y.J.; writing—original draft preparation, X.-G.L. and C.-C.C.; writing—review and editing, Y.-H.F.; visualization, Y.-H.F.; supervision, Y.-H.F.; project administration, L.-H.W. and C.-C.C.; funding acquisition, X.-X.Y.

**Funding:** This research was funded by the Natural Science Foundation of Shanxi Province, China (201801D221118), the National Natural Science Foundation of China (grant nos. 11802194 and 11602157) and the Taiyuan University of Science and Technology Scientific Research Initial Funding (TYUST SRIF. 20152027, 20162037).

**Conflicts of Interest:** The authors declare no conflict of interest. We confirm that the manuscript has been read and approved by all named authors and that there are no other persons who satisfied the criteria for authorship but are not listed. We further confirm that the order of authors listed in the manuscript has been approved by all of us. We confirm that we have given due consideration to the protection of intellectual property associated with this work and that there are no impediments to publication, including the timing of publication, with respect to intellectual property. In so doing we confirm that we have followed the regulations of our institutions concerning intellectual property.

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
