# Peer review of "Investigation of High-Efficiency Iterative ILU Preconditioner Algorithm for Partial-Differential Equation Systems"

_symmetry, doi:10.3390/sym11121461_

Round 1

Reviewer 1 Report

The paper typesetting is very bad, and I guess that it is due to the improper tool used for editing. In some lines, A is small in some large. So it is tough to read the manuscript. It reminds me of a 50 years paper which was written on a typewriter.

The algorithm itself is pretty standard, it is a Richardson method with an ILU preconditioner. The only real  novelty is in the parallel tests. However the 1st example has nothing to do with any PDE, and it is not said what discretization of example 2 is used in the tests? Finite Difference? Finite Element? It should be briefly explained.

The authors compare their numerical results with some other parallel methods, but a reader may not know what they are, so I think a brief description of those methods is necessary.  It should be stressed out where the main differences lie between those methods and the new one of this paper.

Author Response

Journal: Symmetry (ISSN 2073-8994)

Manuscript ID: symmetry-628941

Type: Article   Number of Pages: 18

Title: Investigation of High Efficiency Iterative ILU Preconditioner Algorithm for

Partial-Differential Equation Systems

Dear Editor,

Thank you very much for your letter and for the comments by the reviewers. These comments are very valuable and helpful for our paper.

We appreciate the careful, constructive, and generally favorable reviews given to our paper by the reviewers.

We believe we have adequately addressed all the excellent advices and questions raised by reviewers. Furthermore we checked the manuscript and made sure the submitted manuscript is correct.

Please contact us if any further questions remain.

Sincerely yours,

Prof.  Chih-Cheng Chen

Response to the comments of reviewers:

Reviewer 1:

Comments and Suggestions for Authors

Q:Thanks for the editor's suggests.

The paper typesetting is very bad, and I guess that it is due to the improper tool used for editing. In some lines, A is small in some large. So it is tough to read the manuscript. It reminds me of a 50 years paper which was written on a typewriter.

The algorithm itself is pretty standard, it is a Richardson method with an ILU preconditioner. The only real  novelty is in the parallel tests. However the 1st example has nothing to do with any PDE, and it is not said what discretization of example 2 is used in the tests? Finite Difference? Finite Element? It should be briefly explained.

The authors compare their numerical results with some other parallel methods, but a reader may not know what they are, so I think a brief description of those methods is necessary.  It should be stressed out where the main differences lie between those methods and the new one of this paper.

Ans: Thanks for the editor's suggests.

Sorry, our tool for editing is improper somewhere. We reviseas follows: We modify  to bold type (line 53). We edit ,,, and so on to bold and italic type (line 68). We edit the coefficient matrix A to bold and italic type (line 70). We revise the structure GH to bold and italic type (line 101). We edit the structure G and H to bold and italic type (line 102). We edit the matrix A to bold and italic type (line 111). We edit the matrix ,and, (line 122). We change the M-matrix A larger (line 127). We edit  to bold (line 129). We revise two m in m-th to italic type (line 130). We edit  and  larger (line 130). We edit  larger (line 132). We revise two m in m-th to italic type (line 135). We edit  and  larger (line 135). We edit  larger (line 136). We edit  larger (line 137). We change the two matrix A to italic type and larger (line 140). We edit larger (line 141). We edit the matrix A larger (line 142). We edit  larger (line 144). We edit  larger (line 146). We edit  larger (line 152). We edit the matrix A larger (line 176). We revise the letter i in the ith to italic type (line 180). We edit and  larger (line 184). We edit  larger (line 187). We edit  larger (line 188). We revise the letter i in the ith to italic type (line 189). We edit the coefficient matrix A to bold and italic type (line 206). The 1st example is not the numerical simulation about any PDE, we use this example in order to test the correctness of the iterative incomplete lower and upper factorization preconditioner algorithm. The first example can build good foundation for the second example about PED. 

The finite difference method is used to discretize example 2 in the tests. We adopt the second-order central difference schemes to discretize the example 2, then we converse the format for numerical simulation, lastly, we test the iterative incomplete lower and upper factorization preconditioner algorithm on different processors.

Wecompare the numerical results with some other parallel methods, such as the BSOR method, the PEk technique and the MPA algorithm.

We add: The multi-splitting algorithm (MPA) [2] can be used to solve large band linear systems of equations, however, it sometimes has lower parallel efficiency (from line 32 to line 34).

We add: The block successive over relaxation method (BSOR) [10] can be adopted to solve the large-scale system of equations, but different parallel efficiency will be presented because of the different optimal relaxation factors (from line 39 to line 41). 

We add: The Pseudo-Elimination method with parameter k (PEk) [40] can be apply on multi-processors, however, the setting of parameter k will influent the speedup and parallel efficiency (from line 46 to line 48).  

The main differences between those methods and the new one of this paper are the new algorithm requires less multiplication and add calculation among every iteration, so this algorithm has more speedup and higher parallel efficiency.

In the introduction, we revise as follows: 1) we add: These algorithms use parallelism to solve banded equations, but they can’t contain solving partial differential equations (from line 42 to line 43). 2) we add: They are mostly preconditioner for sparse linear systems or partial differential equations problems in GPU computation (from line 48 to line 49).3) we add: It requires the development of new algorithm, who need less calculation among every iteration and has more speedup and higher parallel efficiency. So that, this paper is concerned with partial-differential equation systems of the form ( from line 50 to line 53). Firstly, we describe a decomposition strategy of parallel algorithm—our proposed ILUP algorithm. Then we analyze the convergence for ILUP algorithm and perform ILUP algorithm. Finally, we compute with numerical examples including the large-scale system of equations and the partial-differential equations and take the conclusions. We replace «Therefore, the new iterative scheme for the block-tridiagonal linear system of equations is »with « Therefore, the new iterative scheme for the large-scale band system of equations is» (from line 94 to line 95). We add «The strategy is as an iterative incomplete lower and upper (ILU) factorization preconditioner (ILUP)algorithm. Compared with the published algorithm [2][10][40], ILUP algorithm requires less multiplication and add calculation among every iteration, so this algorithm has more speedup and higher parallel efficiency. It is appropriate for solving the large-scale system of equations and the partial-differential equations on the multi-core processors» (from line 103 to line 107). We replace «NA » with « ILUP » (line 407). We replace « proposed NA » with « ILUP » (line 412). We replace « the proposed NA » with « ILUP » (line 415). We replace « our NA » with « ILUP » (line 417). We replace « the NA » with « ILUP » (line 419). We replace « our proposed NA » with « ILUP » (line 420). We replace « the NA » with « ILUP » (line 421). We replace « the NA algorithm presented in this paper » with « ILUP algorithm » (line 422). We replace « the NA » with « ILUP » (line 425).

Reviewer 2 Report

The authors presented an interesting iterative algorithm that improves the efficiency of parallel computing.

Some minor bugs:

- An extra dot in the header of section 3.2 (line 101);

- The authors use the abbreviation «NA» to denote the proposed algorithm. However, this designation is first used in line 187 without interpretation. The reader can only guess that this may be the «New Algorithm»? I suggest introducing the name of the developed algorithm and using this name in the paper instead of the long and constantly repeating phrases «… our proposed NA algorithm …» and «… the NA algorithm in this paper …»

- Extra comma and quotation mark on line 445 ... Ma, “Parallel ...

- An extra dot on line 459. It should be: «Terekhov, A.V. A fast parallel algorithm ...»

Overall recommendation: Accept after minor revision (text editing).

Author Response

Journal: Symmetry (ISSN 2073-8994)

Manuscript ID: symmetry-628941

Type: Article   Number of Pages: 18

Title: Investigation of High Efficiency Iterative ILU Preconditioner Algorithm for Partial-Differential Equation Systems

Dear Editor,

Thank you very much for your letter and for the comments by the reviewers. These comments are very valuable and helpful for our paper.

We appreciate the careful, constructive, and generally favorable reviews given to our paper by the reviewers.

We believe we have adequately addressed all the excellent advices and questions raised by reviewers. Furthermore we checked the manuscript and made sure the submitted manuscript is correct.

Please contact us if any further questions remain.

Sincerely yours,

Chih-Cheng Chen

Response to the comments of reviewers:

Reviewer 2:

Comments and Suggestions for Authors

The authors presented an interesting iterative algorithm that improves the efficiency of parallel computing.

Some minor bugs:

Q1:

- An extra dot in the header of section 3.2 (line 101);

- The authors use the abbreviation «NA» to denote the proposed algorithm. However, this designation is first used in line 187 without interpretation. The reader can only guess that this may be the «New Algorithm»? I suggest introducing the name of the developed algorithm and using this name in the paper instead of the long and constantly repeating phrases «… our proposed NA algorithm …» and «… the NA algorithm in this paper …»

- Extra comma and quotation mark on line 445 ... Ma, “Parallel ...

- An extra dot on line 459. It should be: «Terekhov, A.V. A fast parallel algorithm ...»

Overall recommendation: Accept after minor revision (text editing).

Ans:  Thanks for the editor's suggests.

We delete the dot at the last of section 3.2 Proposition and theoremWith our proposed algorithm, we replace the abbreviation «NA» with the abbreviation «ILUP». This designation is firstly used in line 104 with interpretation. We replace NA with ILUP on line We replace « the new algorithm produced in this paper » with «the iterative incomplete lower and upper factorization preconditioner» (line 203). We replace « our proposed ILU factorization preconditioner for the large-scale system of equations (NA) » with «the iterative incomplete lower and upper factorization preconditioner (ILUP)» (from line 215 to line 216). We replace « The solutions to the algorithm presented in the paper (NA)» with «the iterative incomplete lower and upper factorization preconditioner (ILUP)» (line 225). We replace « our proposed ILU factorization preconditioner» with « ILUP algorithm» (line 233). We replace « our proposed algorithm » with « ILUP algorithm» (line 234). We replace « the algorithm proposed ILU factorization preconditioner » with « ILUP algorithm» (line 234). We replace « our proposed algorithm in this paper (NA)» with « ILUP algorithm» (line 237). We replace « our proposed algorithm (NA) in this paper » with « ILUP algorithm» (line 238). We replace « our proposed ILU factorization preconditioner » with « ILUP algorithm» (line 247). We replace « our proposed algorithm » with « ILUP algorithm» (line 248). We replace « the algorithm proposed ILU factorization preconditioner » with « ILUP algorithm» (line 249). We replace « the NA algorithm in this paper » with « ILUP algorithm» (line 251). We replace « the NA » with « ILUP » (line 253).

We replace « our proposed ILU factorization preconditioner » with « ILUP algorithm» (line 261). We replace « our proposed NA algorithm » with « ILUP algorithm» (line 262). We replace « the NA algorithm proposed » with « ILUP algorithm» (line 263). We replace « the NA algorithm in this paper » with « ILUP algorithm» (line 264). We replace « the NA » with « ILUP » (line 266). We replace « our proposed NA algorithm » with « ILUP algorithm» (line 269). We replace « the NA » with « ILUP » (line 274). We replace « new algorithm produced in this paper (NA) » with « ILUP algorithm » (row1 column 2 in Table 5). We replace « our proposed NA algorithm » with « ILUP algorithm » (line 281). We replace « the NA methods » with « ILUP algorithm» (line 284). We replace NA with ILUP in Figure 1 (line 288).

We replace « our proposed NA algorithm » with « ILUP algorithm » (line 290). We replace « the NA methods » with « ILUP algorithm» (line 293). We replace « the NA methods » with « ILUP algorithm» (line 297). We replace « the NA » with « ILUP » (line 299). We replace NA with ILUP in Figure 2 (line 300).

We replace « our proposed ILU factorization preconditioner for the partial-differential equations (NA) » with «the iterative incomplete lower and upper factorization preconditioner (ILUP)» ( line 316). We replace « The solutions to the algorithm presented in the paper (NA)» with «the iterative incomplete lower and upper factorization preconditioner (ILUP)» (line 324). We replace « our proposed ILU factorization preconditioner» with « ILUP algorithm» (line 331). We replace « our proposed algorithm » with « ILUP algorithm» (line 332). We replace « the algorithm proposed » with « ILUP algorithm» (line 334-335). We replace « our proposed algorithm in this paper» with « ILUP algorithm» (line 336). We replace « our proposed ILU factorization preconditioner » with « ILUP algorithm» (line 344). We replace « the NA algorithm» with « ILUP algorithm» (line 345). We replace « the NA algorithm proposed in this paper » with « ILUP algorithm» (line 345). We replace « the NA algorithm in this paper » with « ILUP algorithm» (line 347). We replace « the NA algorithm in this paper » with « ILUP algorithm» (line 348-349). We replace « our proposed ILU factorization preconditioner » with « ILUP algorithm» (line 356-357). We replace « our proposed NA algorithm » with « ILUP algorithm» (line 358). We replace « the NA algorithm » with « ILUP algorithm» (line 358). We replace « the NA algorithm in this paper » with « ILUP algorithm» (line 360). We replace « the NA » with « ILUP » (line 363). We replace « our proposed NA algorithm » with « ILUP algorithm» (line 366). We replace « the NA » with « ILUP » (line 371). We replace « new algorithm produced in this paper (NA) » with « ILUP algorithm » (row1 column 2 in Table 10). We replace « our proposed NA algorithm » with « ILUP algorithm » (line 377). We replace « the NA methods » with « ILUP algorithm» (line 380). We replace « the NA methods » with « ILUP algorithm» (line 384). We replace « the NA » with « ILUP » (line 386). We replace NA with ILUP in Figure 3 (line 387).

We replace « our proposed NA algorithm » with « ILUP algorithm » (line 391). We replace « the NA methods » with « ILUP algorithm» (line 394). We replace « the NA methods » with « ILUP algorithm» (line 398). We replace « the NA » with « ILUP » (line 400). We replace NA with ILUP in Figure 4 (line 402).

Q2: Extra comma and quotation mark on line 445 ... Ma, “Parallel ...

Ans:  Thanks for the editor's suggests.

We delete thecomma and quotation mark on line 458 in the current version. Replace «... Ma, “Parallel ... » with «... Parallel ... »

Q3:An extra dot on line 459. It should be: «Terekhov, A.V. A fast parallel algorithm ..»

Ans:  Thanks for the editor's suggests.

We delete theextra dot on line 472 in the current version. Replace «Terekhov, A. A. fast-parallel algorithm ...» with «Terekhov, A.V. A fast parallel algorithm ...»
